# Influenza Vaccination Uptake in the General Italian Population during the 2020–2021 Flu Season: Data from the EPICOVID-19 Online Web-Based Survey

**DOI:** 10.3390/vaccines10020293

**Published:** 2022-02-15

**Authors:** Andrea Giacomelli, Massimo Galli, Stefania Maggi, Marianna Noale, Caterina Trevisan, Gabriele Pagani, Raffaele Antonelli-Incalzi, Sabrina Molinaro, Luca Bastiani, Liliana Cori, Fabrizio Bianchi, Nithiya Jesuthasan, Federica Prinelli, Fulvio Adorni

**Affiliations:** 1III Infectious Diseases Unit, ASST Fatebenefratelli Sacco, Via G.B. Grassi 74, 20157 Milan, Italy; massimo.galli@unimi.it; 2Dipartimento di Scienze Biomediche e Cliniche Luigi Sacco, Università degli Studi di Milano, Via G.B. Grassi 74, 20157 Milan, Italy; 3CNR Aging Branch-NI, Via Giustiniani 2, 35128 Padua, Italy; stefania.maggi@in.cnr.it (S.M.); marianna.noale@in.cnr.it (M.N.); 4Dipartimento di Scienze Mediche, Università di Ferrara, Via Fossato di Mortara 64, 44121 Ferrara, Italy; caterina.trevisan.5@studenti.unipd.it; 5Malattie Infettive, Ospedale Nuovo di Legnano, ASST Ovest Milanese, Via Papa Giovanni Paolo II, 20025 Legnano, Italy; gabriele.pagani@unimi.it; 6Unit of Geriatrics, Department of Medicine, Biomedical Campus of Rome, Via Alvaro del Portillo 21, 00128 Rome, Italy; r.antonelli@unicampus.it; 7Epidemiology and Health Research Laboratory, National Research Council, Institute of Clinical Physiology, Via G. Moruzzi 1, 56124 Pisa, Italy; sabrina.molinaro@ifc.cnr.it (S.M.); lucabastiani@ifc.cnr.it (L.B.); 8Department of Environmental Epidemiology and Disease Registries, National Research Council, Institute of Clinical Physiology, 56124 Pisa, Italy; liliana.cori@ifc.cnr.it (L.C.); fabriepi@ifc.cnr.it (F.B.); 9National Research Council, Institute of Biomedical Technologies, Via Fratelli Cervi 93, 20054 Segrate, Italy; nithiya.jesuthasan@itb.cnr.it (N.J.); federica.prinelli@itb.cnr.it (F.P.); fulvio.adorni@itb.cnr.it (F.A.)

**Keywords:** flu vaccination, COVID-19, SARS-CoV-2, vaccine hesitancy, determinants, influenza season

## Abstract

To assess influenza vaccine uptake during the 2020/2021 flu season and compare it with that of the 2019/2020 flu season among respondents to the second phase of the web-based EPICOVID-19 survey, we performed an observational web-based nationwide online survey (January–February 2021) in which respondents to the first survey (April–June 2020) were contacted and asked to complete a second questionnaire. Factors associated with vaccine uptake in the 2020/2021 flu season were assessed by applying a multivariable multinomial logistic regression model. Out of the 198,822 respondents to the first survey, 41,473 (20.9%) agreed to fill out the follow-up questionnaire; of these, 8339 (20.1%) were vaccinated only during the 2020/2021 season, 8828 (21.3%) were vaccinated during both seasons and 22,710 (54.8%) were vaccinated in neither season. Educational level (medium (aOR 1.33 95%CI 1.13–1.56) and high (aOR 1.69 95%CI 1.44–1.97) vs. low) and socio-economic deprivation according to SES scoring (1 point aOR 0.83 (95%CI 0.78–0.89), 2 aOR 0.68 (95%CI 0.60–0.77) points or ≥3 points aOR 0.42 (95%CI 0.28–0.45) vs. 0 points) were found to be associated with flu vaccine uptake. Our study shows that social determinants seemed to affect flu vaccination uptake and identifies specific categories of the population to target during future influenza vaccination campaigns.

## 1. Introduction

Influenza vaccination is a major public health intervention that is able to reduce the influenza burden both at the individual and population levels [1,2,3,4]. The impact of different flu seasons on morbidity and excess mortality is variable and, for Italy, there are still gaps in existing data on this topic. Nevertheless, there is evidence of the significant burden that influenza places each year, especially on high-risk groups [5]. In addition, Italy is particularly prone to be affected by the threat of influenza considering the composition of its population with a significant proportion of people aged above 75 years, including a large proportion of frail people [6]. Vaccine hesitancy, which seems to be a worldwide phenomenon, has been undermining the efforts of healthcare officials and health professionals to reach an adequate flu vaccination coverage in target categories of the population [7].

A vaccination coverage that was much lower than the 75% coverage of target groups recommended by the World Health Organization has been registered in Italy across several flu seasons [8]. In particular, data from the national surveillance system showed a progressive increase in flu vaccine uptake in Italy until the 2009/2010 flu season, at which time the vaccination coverage for the overall population was 19.6% per 100 inhabitants; until 2008/2009 there was a vaccination coverage for those aged 65 years or older of 66.3%. Thereafter, a progressive decline was subsequently observed for those aged 65 years or older, until a coverage of 48.5% was registered in 2014/2015, followed by a slight progressive increase over the following years [9]. According to the official Italian Ministry of Health data regarding vaccination coverage during the 2019/2020 flu season, the overall vaccination coverage in the whole Italian population was 16.7% and 54.6% in those aged 65 years or older, confirming the trend toward an increasing vaccination coverage, although not yet optimal [10].

The 2020/2021 flu season was faced with unprecedented challenges due to the potential collision of the SARS-CoV-2 pandemic and seasonal flu epidemic [11]. Although the 2020/2021 flu season was almost entirely suppressed in the northern hemisphere [12], preliminary data gathered during the early days of the COVID-19 pandemic suggested, on the one hand, an increased risk of death in the event of influenza and SARS-CoV-2 coinfection [13] while, on the other, a lower risk of severe COVID-19 disease course in subjects who underwent flu and/or pneumococcal vaccinations during the 2019/2020 season [14,15,16,17]. Nevertheless, instead of reinforcing the message of the extensive clinical benefit of influenza vaccination, the extension of the free vaccination to those aged 60–64 years was justified by a theoretical advantage of flu vaccination in the differential diagnosis of influenza-like illness (ILI) in this age group [18]. Consequently, health authorities extended the eligible population for the flu vaccine, i.e., in Italy, it was extended free of charge to all subjects aged ≥60 in addition to the “at-risk” population [19]. Thanks to the 2020/2021 flu vaccine campaign, in Italy there was a significant increase in vaccination coverage in the general population (from 16.8% in 2019/2020 to 23.7% in 2020/2021) and an even higher increment in older adults (from 54.6% to 65.3%, respectively) [10]. This increase was observed in all European countries despite the rapid shortage of flu vaccine stocks observed in the European continent during the first weeks of the influenza season 2020/2021 [20,21,22].

While the determinants of the higher vaccination rates during the 2020/2021 flu season are yet to be clearly understood, clinicians and public health officials are seeking to uncover what categories of the population are not being reached or are being left behind by information campaigns to educate patients about the benefit of the flu vaccine during the COVID-19 pandemic period. Therefore, the aim of our study was to assess and compare influenza vaccine uptake during the 2019/2020 and 2020/2021 flu seasons in a self-selected sample of respondents to the web-based EPICOVID-19 survey and to identify the factors associated with getting an influenza vaccination.

## 2. Materials and Methods

### 2.1. Study Design and Setting

The study design and methodology are described in detail elsewhere [23]. Briefly, the EPICOVID-19 was a two-phase internet-based survey performed in a self-selected sample of adult volunteers aged 18 or older living in Italy during the first and second waves of the pandemic. During the first phase (from 13 April to 2 June 2020), the participants were recruited via social media, press releases, internet pages, local radio, TV stations, word of mouth and the study website (https://epicovid19.itb.cnr.it/ accessed on 26 October 2021). The inclusion criteria were age ≥18 years, access to a mobile phone, computer or tablet with internet connectivity and the provision of online consent to participate in the study. Participants who consented to be contacted by providing their personal e-mail address during the first phase (*n* = 105,355) received an email invitation containing a personalised link to complete the second questionnaire (from 15 January to 28 February 2021). After excluding the individuals who did not receive the invitation to participate, who did not respond (*n* = 63,203), who did not provide consent (*n* = 653) and those with incongruences in the email contacts or who answered more than once using the same email address (*n* = 26), we were left with 41,473 individuals who agreed to participate in the second phase of the study.

### 2.2. Data Collection and Variables

For the purpose of the study, the participants were categorized according to the influenza vaccine uptake during the 2019/2020 and 2020/2021 flu seasons and classified as: vaccinated in neither one of the seasons, vaccinated in both seasons, vaccinated in the 2019/2020 season only or vaccinated in 2020/2021 season only.

The following variables were also considered: biological sex (female vs. male); age (18–59, 60–64, 65 or more); education (categorized as low (primary school or less), medium (middle or high school) or high (university degree or post-graduate)); employment and work category at risk of infection (categorized as employed at a job not at risk, employed as school staff, employed as health care staff, other at-risk employment, unemployed, students, retired or other); smoking habit (no, former or current); and alcohol consumption (never, ≤once a week, 2–3 times a week, 4 or more times a week).

A set of proxy variables (dichotomized) were decided upon and assessed as a measure of the participant’s socio-economic status (SES). These were employment vs. unemployment, ownership vs. non-ownership of home, ownership vs. non-ownership of a car and a congruous number of people living in the house vs. house-crowding (defined as a number of cohabitants greater than the number of rooms in the house, excluding the kitchen and bathrooms) [24]. The total score ranged from 0 to 4, with a higher score indicating a lower socio-economic status. Geographical areas of residence were classified as Italian regions and others. As far as participants’ health status was concerned, the participants’ body mass index (BMI), calculated as weight divided by height squared, was further categorized as: healthy weight (18.5–24), underweight (<18.5), overweight (25–29), obesity (≥30) or unknown. The number of comorbidities was attained by summing the participants’ chronic conditions (lung diseases, heart diseases, hypertension, renal diseases, diseases of the immune system, oncological diseases, metabolic diseases, neurological diseases, cerebrovascular diseases, liver diseases (hepatitis C, B and other disorders), depression and/or anxiety, eating disorders and anemia) and then categorizing the results as: none, 1, 2 or 3 or more. Another variable that was taken into consideration was whether the respondent had had a positive result for a COVID-19 test (a nasopharyngeal swab (NPS) or a serological test (ST)) during the study period.

### 2.3. Statistical Analysis

The sample characteristics are expressed as mean and standard deviation (SD) for continuous variables and as count and percentages for the categorical variables. Comparison of the participants’ characteristics according to influenza vaccine uptake was performed with a chi-squared test. Flu vaccine rates were calculated for each age class and for every Italian region. The factors associated with vaccine uptake in the 2019/2020 and 2020/2021 flu seasons were assessed by applying a multivariable multinomial logistic regression model considering those who were not vaccinated during neither flu season as the reference category. A set of a priori arbitrary chosen variables which have been considered to be potentially related to influenza vaccination uptake were entered and adjusted for in the final multivariable multinomial model (age, biological sex, education level, type of employment, COVID-19 during the study period, number of morbidities, smoking habits, alcohol drinking between meals, SES and BMI). Adjusted odds ratios (aOR) and 95% confidence intervals (CIs) were estimated for each independent variable. We further performed a sensitivity analysis stratifying by age group (participants aged <65 years and aged ≥65 years). Two-tail *p*-values of statistical significance were set at 0.05. All of the statistical analyses were carried out using SPSS (IBM Corp. Released, IBM SPSS Statistics version 25.0 Armonk, NY, USA: IBM Corp.).

### 2.4. Ethics and Consent Form

The Ethics Committee of the Istituto Nazionale per le Malattie Infettive (the National Institute for Infectious Diseases) IRCCS Lazzaro Spallanzani approved the protocol for the first phase of the EPICOVID-19 study (No. 70, 12 April 2020) and later the protocol for the second phase (No. 249, 14 January 2021). The participants were requested to give their informed consent when they first accessed the online platform. Participation was voluntary and no compensation was given to respondents. The planning, conduct, and reporting of the study were in line with the Declaration of Helsinki, as revised in 2013. The data were handled and stored in accordance with the European Union General Data Protection Regulation (EU GDPR) 2016/679; data transfer included encryption/decryption and password protection.

## 3. Results

### 3.1. Characteristics of Respondents

Out of the 198,822 respondents to the first survey, 41,473 (20.9%) accepted to respond to the follow-up questionnaire; of these, 8,828 (21.3%) said they had been vaccinated during both the 2019/2020 and the 2020/2021 flu seasons, while 1596 (3.8%) were vaccinated only during the 2019/2020 season, 8339 (20.1%) were vaccinated only during the 2020/2021 season and 22,710 (54,8%) said they had been vaccinated in neither year.

The characteristics of the study population according to vaccine uptake in the 2019/2020 and 2020/2021 flu seasons are reported in Table 1.

The mean age of respondents was 50.7 (SD ± 13.5) years, 25,146 (60.6%) were females, 27,158 (65.5%) reported a high education level, 26,124 (62.3%) reported that they had a stable work position. Among respondents, 26,029 (62.8%) reported to have no comorbidities. The majority of respondents were from the Lombardia region (13,832, 33.3%), followed by Piemonte (4644, 11.2%), Emilia-Romagna (4485, 10.8%) and Lazio regions (4072, 9.8%).

### 3.2. Vaccination Coverage According to the Different Italian Regions

Vaccination coverage in 2019/2020 and 2020/2021 seasons in the different Italian regions is reported in Appendix A. Respondents from Valle d’Aosta, Abruzzo and Piemonte were those with the highest frequency of no vaccine uptake in both 2019/2020 and 2020/2021 flu seasons (65.8%, 63.9% and 61.4%, respectively). Respondents from Lazio, Puglia and Campania reported the highest frequency of vaccine uptake only in the 2020/2021 flu season (27.32%, 24.8% and 23.5%, respectively).

The Molise, Sicilia, Emilia-Romagna and Friuli-Venezia Giulia regions had the highest number of respondents who underwent flu vaccination during both the 2019/2020 and 2020/2021 flu seasons (27%, 24.4, 24.3% and 24.1%, respectively).

### 3.3. Factors Associated with Flu Vaccine Uptake during the 2019/2020 and 2020/2021 Flu Seasons

Factors associated with vaccine uptake in the 2019/2020 and 2020/2021 flu seasons are reported in Table 2.

Age (60–64 years aOR 2.72 (95%CI 2.50–2.96) and ≥65 years aOR 4.61 (95%CI 4.07–5.23) vs. 18–59 years), education level (medium (aOR 1.33 95%CI 1.13–1.56) or high (aOR 1.69 95%CI 1.44–1.97) vs. low), number of morbidities (1 aOR 1.32 (95%CI 1.24–1.41), 2 aOR 1.51 (95%CI 1.37–1.66) or ≥3 aOR 1.83 (95%CI 1.61–2.08) vs. none), being a health care worker (vs. employment at a low risk job aOR 2.66 95%CI 2.42–2.92) results associated with higher odds of influenza vaccine uptake during the 2020/2021 flu season only. On the contrary, lower odds were observed in those with a higher deprivation according to SES score (1 point aOR 0.83 (95%CI 0.78–0.89), 2 points aOR0.68 (95%CI 0.60–0.77) or ≥3 points aOR 0.42 (95%CI 0.28–0.45) vs. 0 points), in those who drink alcohol between meals 2–3 times a week (vs. never drink aOR 0.89 (95%CI 0.82–0.98)) and in those underweight (aOR 0.84 (95%CI 0.72–0.99)).

By comparing those who received influenza vaccination both in 2019/2020 and 2020/2021 with those who did not received a flu vaccine shot during neither, being male (aOR 1.25 (95%CI 1.18–1.33)), age (60–64 years aOR 2.67 (95%CI 2.44–2.93) or ≥65 years aOR 9.23 (95%CI 8.18–10.42) vs. 18–59 years), education level (high (aOR 1.71 95%CI 1.46–2.00) vs. low), number of morbidities (1 aOR 1.48 (95%CI 1.39–1.58), 2 aOR 2.12 (95%CI 1.93–2.33) or ≥3 aOR 2.94 (95%CI 2.60–3.33) vs. none), being a health care worker (vs. work not at risk aOR 4.49 95%CI 4.08–4.94) were found to be associated with higher odds of being vaccinated. On the contrary, lower odds were observed in smokers (aOR 0.63 (95%CI 0.58–0.68)), in those who drink alcohol between meals (≤1 a week aOR 0.87 (95%CI 0.81–0.94), 2–3 times a week aOR 0.78 (95%CI 0.71–0.85) or ≥4 times a week aOR 0.88 (95%CI 0.80–0.96) vs. never drink), in those underweight (aOR 0.83 (95%CI 0.70–0.99)) and in those with a higher deprivation assessed by SES score (1 point aOR 0.79 (95%CI 0.74–0.85), 2 points aOR0.59 (95%CI 0.51–0.67) point or ≥3 points aOR 0.33 (95%CI 0.21–0.54) vs. 0 points).

The results of the analysis comparing those who received and those who did not receive an influenza vaccine shot during the 2020/2021 flu season stratified by age group are reported in Figure 1 (<65 years) and Figure 2 (≥65 years).

In those aged <65 years (per 1 year more) (aOR 1.03 (95%CI 1.02–1.03)), education level (medium aOR 1.42 (95%CI 1.18–1.72) or high aOR1.87 (95%CI 1.55–2.25) vs. low), being health care staff (aOR 2.91 (95%CI 2.64–3.20)), being retired (aOR 1.95 (95%CI 1.71–2.22)) and number of morbidities (1 aOR 1.34 (95%CI 1.25–1.43) or 2 aOR 1.51 (95%CI 1.36–1.68) vs. ≥3 aOR 1.86 (95%CI 1.61–2.14)) and being overweight (aOR 1.09 (95%CI 1.02–1.16)) were associated with higher odds of being vaccinated during the 2020/2021 flu season. On the contrary, a previous COVID-19 diagnosis (aOR 0.81 (95%CI 0.74–0.89)), being an active smoker (aOR 0.86 (95%CI 0.80–0.93)), drinking alcohol between meals (aOR 0.91 (95%CI 0.83–1.00)) and a higher deprivation according to SES score (1 point aOR 0.91 (95%CI 0.85–0.97), 2 points aOR 0.75 (95%CI 0.66–0.86) or ≥3 points aOR 0.42 (95%CI 0.27–0.67) vs. 0 points) were associated with lower odds of influenza vaccine uptake.

In those aged ≥65 years, a higher education level (vs. low aOR 1.51 (95%CI 1.08–2.10)) was associated with higher probability of receiving an influenza vaccination during the 2020/2021 flu season. On the contrary, a higher deprivation according to SES score (1 point aOR 0.70 (95%CI 0.58–0.86) vs. 0 points) and being underweight (aOR 0.48 (95%CI 0.27–0.85)) were associated with lower odds of influenza vaccine uptake.

## 4. Discussion

This study used the data obtained with a two-phase web-based online survey regarding the COVID-19 pandemic carried out in Italy to analyse and compare the differences in influenza vaccination rates during the 2019/2020 and 2020/2021 flu seasons, a time that corresponds to the ongoing COVID-19 pandemic [11,12,13,14,15,16,17]. The study’s primary result was that influenza vaccine uptake jumped from 25.1% in the 2019/2020 flu season to 41.4% in the 2020/2021 season. In addition, it uncovered several demographic and social variables that seemed to be associated with vaccination rates.

Although not comparable to the data provided by the Italian Ministry of Health on flu vaccination coverage in the general population, the increased proportion of flu coverage observed in our study is in line with the data regarding the 2020/2021 flu season in Italy [10]. In fact, a significant increase in vaccine coverage, up to 23.7% in the general population and in those aged ≥65 years (65.3%), was observed [10], suggesting that the vaccination campaign during the COVID-19 pandemic period reached a wide audience in the general population when compared with the previous flu vaccination campaigns. This can probably be explained by the intense public information campaigns regarding the potential impact of co-infection with flu and COVID-19 in the absence of effective vaccines and treatment for the latter at the start of the 2020/2021 influenza season [25]. In addition, although not directly assessed in the present survey, it could be speculated that COVID-19 fear could have acted as a driver of influenza vaccine uptake. In fact, as it has been shown, greater fear of being infected with SARS-CoV-2 could be observed in vaccinated people (both for influenza and pneumococcal infections), confirming the possible role of fear in promoting preventive attitudes [26]. Nevertheless, the vaccination coverage observed in those aged ≥65 years in our survey was much higher compared to the official Italian data (79.1% vs. 65.3%, respectively) and suggests a likely selection of respondents with a higher health literacy when compared to the Italian general population [9,10]. In particular, although health literacy was not measured in our questionnaire, it could be speculated that higher health literacy could be represented in our sample considering the high proportion of health care workers among respondents (3523 (8.5%)) [27,28]. This finding highlights a possible selection bias of health care workers. In fact, according to the 2019 ISTAT estimates, there were 753,836 health care workers in Italy out of 59,730,000 residents, accounting for 1.26% of the Italian resident population [29]. In addition, it was likely that the respondents with a higher education level and possibly a consequent higher health literacy were more prone to complete the second survey, considering that a high education level was overrepresented among those who agreed to participate in the second survey compared to those who did not (27,158 (65.5%) vs. 91,010 (57.8%), *p* < 0.0001, respectively) [23].

Our estimates of influenza vaccine uptake were quite comparable to those reported in another survey conducted in two metropolitan cities in Italy during the second SARS-CoV-2 wave, in which 47.3% of respondents were prone to receive the influenza vaccine in the 2020/2021 flu season [30]. In another longitudinal online survey conducted in Italy, Domnich and co-workers found that the willingness to undergo influenza vaccination increased from 44.1% for the 2020/2021 flu season to 48.6% for the 2021/2022 flu season. In addition, previous influenza vaccinations, receipt of a COVID-19 vaccine, positive attitudes towards influenza vaccination, male sex at birth and older age were all associated with willingness to receive the 2021/2022 influenza vaccine [31].

Although the determinants of flu vaccine uptake have been extensively explored during several flu seasons [32,33,34], whether the same determinant affected flu vaccine uptake during the COVID-19 pandemic is still a matter of debate [31,35]. By using data from the first EPICOVID-19 survey, our group highlighted that education level and stable employment was strongly associated with flu vaccine uptake during the 2019/2020 flu season in Italy [36]. With the present analysis, conducted in a subsample of respondents to the first study, we highlighted, again, how a higher education level affects flu vaccine uptake. This is in line with a study conducted in Italy by Damiani and co-workers in the 1999/2000 flu season during which low levels of education and a manual occupation were found to correspond to fewer influenza vaccinations compared to those received by higher educated, white-collar counterparts, with an OR ranging from 0.65 (95%CI 0.55–0.77) [37]. Nevertheless, the effect of specific socioeconomic determinants is not the same in different countries and in different flu seasons, as highlighted by a study conducted in seven consecutive influenza seasons (2001–2008) in 11 European countries. In the same study, immunized respondents were on average more highly educated in Austria and Poland (OR tertiary versus primary: 1.67, *p* = 0.001, and 2.74, *p* < 0.001, respectively). On the contrary, data from Ireland, Italy and Spain showed that education above primary level was associated with lower chances of flu vaccination uptake [38]. The inverted trend toward vaccine uptake in Italy according to education level observed in our study could be explained, on the one hand, by the evolving and dynamic nature of vaccine hesitancy [39] and, on the other, by the potential advantage of more highly educated respondents when it comes to interpreting information regarding vaccines within the context of a pandemic [35,40]. Although digital health literacy and different information seeking behaviours [41] have not been explored in our questionnaire, it could be speculated that changes in digital health literacy during the last decade might have resulted in—in highly educated individuals—better knowledge about influenza vaccination and therefore led to a higher uptake. Other evidence instead suggests that disadvantaged individuals are less likely to get access to vaccinations in general and to the influenza vaccine in particular [32]. The finding could be partially explained by the inability to understand the information regarding the benefit of flu vaccination [42,43]. In line with the data available in the literature, we found that SES score as an index of deprivation was associated with lower odds of influenza vaccine uptake, and the finding confirms the importance of socio-economic determinants in vaccine hesitancy.

It is well established that obese patients are more likely to receive flu and pneumococcal vaccinations [44] and a higher body weight was found to increase the likelihood of being vaccinated for influenza in patients with chronic obstructive diseases in Spain [45]. No convincing evidence linking overweight and vaccination uptake emerged, however, during the analysis of our data. The inverse association in those aged >65 years between being underweight and getting vaccinated could hypothetically be explained if being underweight can be considered a proxy of frailty, as it may in the elderly.

The inverse association observed between being an active smoker and influenza vaccine uptake should be carefully interpreted considering that no definitive evidence is available regarding this topic, with some studies showing a negative impact of smoking on influenza vaccine uptake and other studies a promoting effect [32]. It could therefore be speculated that unmeasured residual confounders could partially explain this association.

The positive association between an increased burden of morbidities and influenza vaccine uptake, especially for those aged <65 years, is not unexpected and is in line with the vaccination campaign during the COVID-19 pandemic period focusing on protecting more vulnerable populations.

Finally, the finding of lower odds of getting an influenza vaccine uptake among those who were diagnosed with COVID-19 during the study period was quite surprising and was confirmed in the analysis to be restricted to those aged <65 years. If, on the one hand, there is some epidemiological evidence of an inverse association between influenza vaccination and SARS-CoV-2 infection [14,15,16,17], on the other, it is possible that other unmeasured residual confounders (for example, behavioural ones) could determine this association.

### Study Limitations and Strengths

Our study has several limitations. First, subjects younger than 18 years were excluded from the survey and consequently the overall data from our study are not comparable to those reported by the Italian Ministry of Health for the general population. Second, the web-based characteristics of the study could have introduced selection biases toward a high proportion of respondents with a higher education level, higher health literacy and higher socioeconomic status, which could partially explain the higher proportion of respondents aged ≥65 years vaccinated for flu in our survey when compared with the Italian general population. In addition, we were not able to assess the direct impact of economic status on flu vaccination uptake, although we used as a proxy in the model education level, working position and the SES. Third, it cannot be excluded that bias toward a better attitude regarding health status and prevention could have been amplified by selecting responders willing to participate in the second point of the survey. Fourth, less than half of respondents to the first online EPICOVID-19 survey were available and allowed to fill out the second questionnaire, thus determining a suboptimal response rate [19]. In the end, influenza virus circulation in Italy was almost suppressed during the 2020/2021 flu season, disregarding the concerns over potential co-circulation of influenza and SARS-CoV-2. In particular, no influenza viruses have been detected by the national surveillance system for the first time in the last twenty-one years, with a seasonal flu epidemic which has not been established in Italy. Although seasonal influenza vaccination coverage has increased compared to previous seasons, several other factors related to COVID-19 containment, such as social physical distancing, an immune memory effect and potential viral competition, could partially explain the suppression of the 2020/2021 flu season [46].

The main strengths of the study are: (1) that it presents two clear pictures of a large number of Italians and their influenza vaccination behavior during two flu seasons corresponding to the ongoing COVID-19 pandemic; and (2) the consistency of the multivariable multinomial logistic regression model applied to those receiving the influenza vaccine in 2020/2021 for the first time and those who reported receiving the vaccine during both the 2019/2020 and 2020/2021 seasons.

## 5. Conclusions

In conclusion, a significantly higher proportion of respondents to the online EPICOVID-19 survey received a flu shot during the 2020/2021 flu season with respect to the 2019/2020 season. The respondents’ educational level and socioeconomic status appeared to be major determinants of flu vaccine uptake during the pandemic period. Although, due to the many changes in the characteristics of pandemic waves, it is not possible to directly transfer the results of the present study to the current situation (fourth wave), some signs that emerged between the second and first waves may be of interest for the present and the future. In particular, the most disadvantaged groups in the population appeared, again, to be left behind and less likely to be reached by the 2020/2021 Italian flu vaccination campaign. Health authorities and policymakers should take these findings into consideration during future influenza vaccine campaigns.

## Figures and Tables

**Figure 1 vaccines-10-00293-f001:**
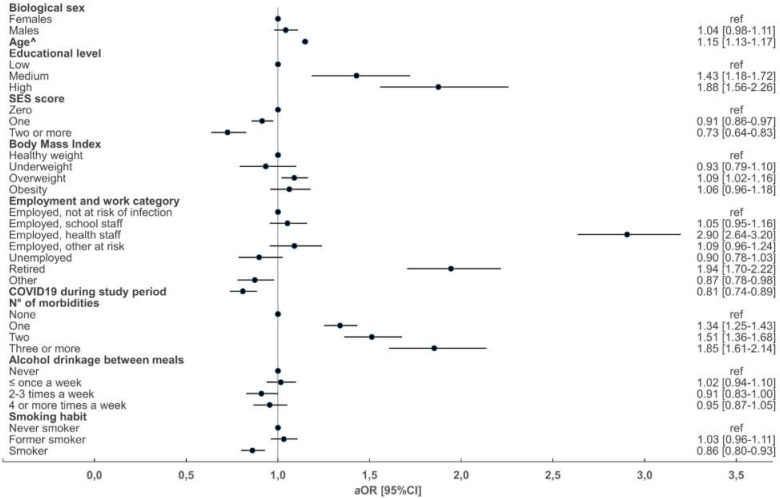
Forest plot of the results from multivariate multinomial logistic regression models of factors associated with flu shot uptake in the 2020/2021 season in respondents aged <65 years. ^ For each additional 5 years of age. ref, reference category; *a*OR, adjusted odds ratios; 95% CI, 95% confidence interval.

**Figure 2 vaccines-10-00293-f002:**
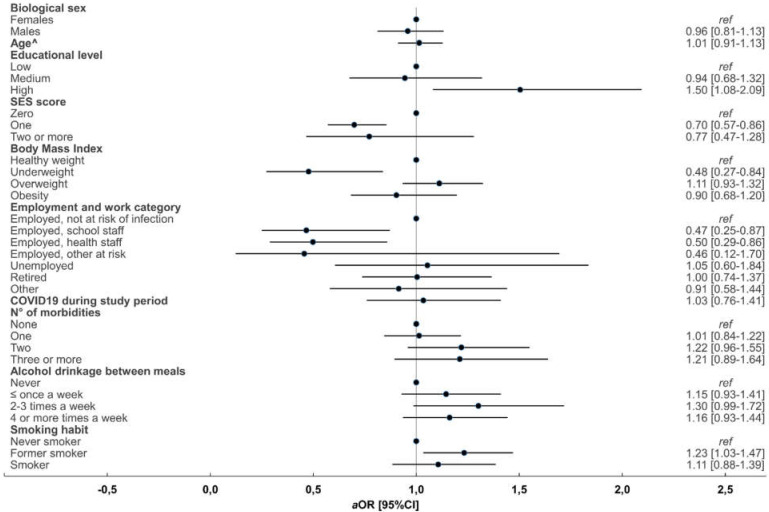
Forest plot of the results from multivariate multinomial logistic regression models of factors associated with flu shot uptake in the 2020/2021 season in respondents aged ≥65 years. ^ For each additional 5 years of age. ref, reference category; *a*OR, adjusted odds ratios; 95% CI, 95% confidence interval.

**Table 1 vaccines-10-00293-t001:** Characteristics of the study population according to vaccine uptake during the 2019/2020 and 2020/2021 flu seasons.

Characteristics	Flu Shot in 2019/2020	NO	YES	NO	YES	Total
	Flu Shot in 2020/2021	NO	NO	YES	YES	
		*n* (%)	*n* (%)	*n* (%)	*n* (%)	
Sex at birth	Male	8513 (37.5)	630 (39.5)	3242 (38.9)	3942 (44.7)	16,327 (39.4)
	Female	14,197 (62.5)	966 (60.5)	5097 (61.1)	4886 (55.3)	25,146 (60.6)
Age class at first interview	18–59	19,776 (87.1)	1375 (86.2)	5240 (62.8)	4090 (46.3)	30,481 (73.5)
	60–64	1857 (8.2)	125 (7.8)	1473 (17.7)	1282 (14.5)	4737 (11.4)
	65+	1077 (4.7)	96 (6.0)	1626 (19.5)	3456 (39.1)	6255 (15.1)
Educational level ^±^	Low	747 (3.3)	44 (2.8)	245 (2.9)	345 (3.9)	1381 (3.3)
	Medium	7479 (32.9)	461 (28.9)	2504 (30.0)	2490 (28.2)	12,934 (31.2)
	High	14,484 (63.8)	1091 (68.4)	5590 (67.0)	5993 (67.9)	27,158 (65.5)
Employment and work category at risk of infection	Employed, not at risk	11,980 (52.8)	806 (50.5)	3409 (40.9)	2378 (26.9)	18,573 (44.8)
	Employed, school staff	2150 (9.5)	163 (10.2)	714 (8.6)	626 (7.1)	3653 (8.8)
	Employed, health staff	1292 (5.7)	121 (7.6)	972 (11.7)	1138 (12.9)	3523 (8.5)
	Employed, other at risk	1228 (5.4)	70 (4.4)	334 (4.0)	206 (2.3)	1838 (4.4)
	Unemployed	1724 (7.6)	101 (6.3)	419 (5.0)	402 (4.6)	2646 (6.4)
	Students	1045 (4.6)	96 (6.0)	147 (1.8)	145 (1.6)	1433 (3.5)
	Retired	1474 (6.5)	125 (7.8)	1835 (22.0)	3457 (39.2)	6891 (16.6)
	Other	1817 (8.0)	114 (7.1)	509 (6.1)	476 (5.4)	2916 (7.0)
COVID-19 ^¶^ during study period		2504 (11.0)	237 (14.8)	811 (9.7)	859 (9.7)	4411 (10.6)
N° of morbidities	None	15,708 (69.2)	1046 (65.5)	4861 (58.3)	4414 (50.0)	26,029 (62.8)
	One	4737 (20.9)	360 (22.6)	2141 (25.7)	2357 (26.7)	9595 (23.1)
	Two	1551 (6.8)	133 (8.3)	860 (10.3)	1261 (14.3)	3805 (9.2)
	Three or more	714 (3.1)	57 (3.6)	477 (5.7)	796 (9.0)	2044 (4.9)
Smoker	No	13,257 (58.4)	1000 (62.7)	4671 (56.0)	4990 (56.5)	23,918 (57.7)
	Former smoker	4818 (21.2)	358 (22.4)	2214 (26.5)	2693 (30.5)	10,083 (24.3)
	Smoker	4635 (20.4)	238 (14.9)	1454 (17.4)	1145 (13.0)	7472 (18.0)
Alcohol drinking between meals	Never	3918 (17.3)	286 (17.9)	1608 (19.3)	1970 (22.3)	7782 (18.8)
	≤once a week	10,329 (45.5)	743 (46.6)	3615 (43.4)	3485 (39.5)	18,172 (43.8)
	2–3 times a week	4526 (19.9)	284 (17.8)	1420 (17.0)	1347 (15.3)	7577 (18.3)
	4 or more times a week	3937 (17.3)	283 (17.7)	1696 (20.3)	2026 (22.9)	7942 (19.1)
SES score *	0	12,591 (55.4)	960 (60.2)	5550 (66.6)	6272 (71.0)	25,373 (61.2)
	1	8133 (35.8)	516 (32.3)	2378 (28.5)	2216 (25.1)	13,243 (31.9)
	2	1781 (7.8)	111 (7.0)	385 (4.6)	320 (3.6)	2597 (6.3)
	3–4	205 (0.9)	9 (0.6)	26 (0.3)	20 (0.2)	260 (0.6)
BMI	Healthy weight (18–24)	13,702 (60.3)	917 (57.5)	4621 (55.4)	4700 (53.2)	23,940 (57.7)
	Underweight (<18.5)	827 (3.6)	77 (4.8)	219 (2.6)	203 (2.3)	1326 (3.2)
	Overweight (25–29)	6069 (26.7)	451 (28.3)	2613 (31.3)	2886 (32.7)	12,019 (29.0)
	Obesity (≥30)	1841 (8.1)	131 (8.2)	765 (9.2)	913 (10.3)	3650 (8.8)
	Unknown	271 (1.2)	20 (1.3)	121 (1.5)	126 (1.4)	538 (1.3)
Total		22,710 (54.8)	1596 (3.8)	8339 (20.1)	8828 (21.3)	41,473 (100)

List of abbreviations: n, number; BMI, body mass index. ^±^ Categorized as low (primary school or less), medium (middle or high school) or high (university degree or post-graduate). * Calculated in accordance with Townsend et al. [20]. ^¶^ Defined as those who reported at least one positive result for a nasopharyngeal swab (NPS) or a serological test (ST) during the considered study period.

**Table 2 vaccines-10-00293-t002:** Factors associated with vaccine uptake in the 2019/2020 and 2020/2021 flu seasons were assessed by applying a multivariable multinomial logistic regression model using those who reported to have not been vaccinated in both seasons as the reference group.

	Flu Shot in 2019/2020	YES		NO		YES	
	Flu Shot in 2020/2021	NO		YES		YES	
Variables		aOR (95% CI)	*p*	aOR (95% CI)	*p*	aOR (95% CI)	*p*
Sex at birth	Females	1	-	1	-	1	-
	Males	1.13 [1.01–1.27]	0.028	1.01 [0.95–1.07]	0.694	1.25 [1.18–1.33]	0.000
Age class at first survey	18–59	1	-	1	-	1	-
	60–64	0.92 [0.74–1.14]	0.447	2.72 [2.50–2.96]	0.000	2.67 [2.44–2.93]	0.000
	65+	1.09 [0.79–1.51]	0.582	4.61 [4.07–5.23]	0.000	9.23 [8.18–10.42]	0.000
Educational level ^±^	Low	1	-	1	-	1	-
	Medium	1.07 [0.78–1.48]	0.659	1.33 [1.13–1.56]	0.001	1.15 [0.98–1.34]	0.089
	High	1.34 [0.98–1.84]	0.070	1.69 [1.44–1.97]	0.000	1.71 [1.46–2.00]	0.000
Employment and work category at risk of infection	Employed, not at risk	1	-	1	-	1	-
	Employed, school staff	1.10 [0.92–1.31]	0.309	1.04 [0.94–1.14]	0.437	1.30 [1.17–1.44]	0.000
	Employed, health staff	1.33 [1.09–1.63]	0.006	2.66 [2.42–2.92]	0.000	4.49 [4.08–4.94]	0.000
	Employed, other at risk	0.89 [0.69–1.15]	0.371	1.07 [0.94–1.22]	0.298	0.99 [0.85–1.16]	0.923
	Unemployed	1.08 [0.85–1.38]	0.510	0.94 [0.83–1.08]	0.387	1.34 [1.16–1.54]	0.000
	Students	1.55 [1.24–1.95]	0.000	0.66 [0.55–0.79]	0.000	1.04 [0.87–1.25]	0.670
	Retired	1.19 [0.88–1.60]	0.264	1.24 [1.10–1.39]	0.000	1.88 [1.67–2.11]	0.000
	Other	0.97 [0.79–1.19]	0.761	0.85 [0.77–0.95]	0.004	1.06 [0.94–1.19]	0.360
COVID19 ^¶^ during study period		1.35 [1.17–1.57]	0.000	0.82 [0.75–0.90]	0.000	0.84 [0.77–0.92]	0.000
N° of morbidities	None	1	-	1	-	1	-
	One	1.14 [1.01–1.30]	0.035	1.32 [1.24–1.41]	0.000	1.48 [1.39–1.58]	0.000
	Two	1.32 [1.09–1.59]	0.005	1.51 [1.37–1.66]	0.000	2.12 [1.93–2.33]	0.000
	Three or more	1.25 [0.94–1.67]	0.117	1.83 [1.61–2.08]	0.000	2.96 [2.62–3.36]	0.000
Smoker	No	1	-	1	-	1	-
	Former smoker	0.99 [0.87–1.13]	0.915	1.04 [0.97–1.11]	0.250	0.98 [0.91–1.04]	0.494
	Smoker	0.71 [0.62–0.83]	0.000	0.88 [0.81–0.94]	0.000	0.63 [0.58–0.68]	0.000
Alcohol drinking between meals	Never	1	-	1	-	1	-
	≤ once a week	0.99 [0.85–1.14]	0.837	1.00 [0.93–1.07]	0.934	0.87 [0.81–0.94]	0.000
	2–3 times a week	0.87 [0.73–1.03]	0.113	0.89 [0.82–0.98]	0.013	0.78 [0.71–0.85]	0.000
	4 or more times a week	1.00 [0.84–1.19]	0.987	0.96 [0.88–1.05]	0.419	0.88 [0.80–0.96]	0.005
SES score *	0	1	-	1	-	1	-
	1	0.82 [0.73–0.93]	0.001	0.83 [0.78–0.89]	0.000	0.79 [0.74–0.85]	0.000
	2	0.81 [0.65–1.02]	0.068	0.68 [0.60–0.77]	0.000	0.59 [0.51–0.67]	0.000
	3–4	0.59 [0.29–1.18]	0.136	0.42 [0.28–0.65]	0.000	0.33 [0.21–0.54]	0.000
BMI	Healthy weight (18.5–24)	1	-	1	-	1	-
	Underweight (<18.5)	1.40 [1.10–1.79]	0.007	0.84 [0.72–0.99]	0.036	0.83 [0.70–0.99]	0.038
	Overweight (25–29)	1.09 [0.97–1.23]	0.152	1.11 [1.05–1.18]	0.001	1.05 [0.98–1.12]	0.151
	Obesity (≥30)	1.03 [0.85–1.25]	0.740	1.08 [0.98–1.19]	0.114	1.09 [0.99–1.21]	0.083
	Unknown	1.10 [0.69–1.75]	0.691	1.17 [0.93–1.47]	0.178	1.06 [0.83–1.35]	0.631

List of abbreviations: CI, confindence interval; aOR, adjusted odds ratio. ^±^ Categorized as low (primary school or less), medium (middle or high school) or high (university degree or post-graduate). * Calculated in accordance with Townsend et al. [20]. ^¶^ Defined as those who reported at least one positive result for a nasopharyngeal swab (NPS) or a serological test (ST) during the considered study period.

## Data Availability

Data were handled and stored following the European Union General Data Protection Regulation (EU GDPR) 2016/679; data transfer included encryption/decryption and password protection. Data will be made available by the authors upon reasonable request.

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
