# Peer review of "Influenza Vaccination Uptake in the General Italian Population during the 2020–2021 Flu Season: Data from the EPICOVID-19 Online Web-Based Survey"

_vaccines, 2022, doi:10.3390/vaccines10020293_

Round 1

Reviewer 1 Report

Dear authors

I’m thankful to review the paper entitled “Influenza Vaccination Uptake in the General Italian Population during the 2020-2021 Flu Season: Data from the EPICOVID 3 Online Web Based Survey” for Vaccines MDPI Journal.

My decision is Accept after minor revision.

The paper is interesting and deal with the emerging topic. The introduction is clear and well arranged. The methodology sounds good and the statistical analyses and study design are readable and informative. The discussion is good even could be improved.

 The findings of the manuscript with a large sample of participants enhance the manuscript.

I suggest enhancing the discussion paragraph debating the outcome of other researches detecting the vaccination coverage in Italy comparing first and second wave pandemic with follow references.

  1. Montalti M, Di Valerio Z, Rallo F, Squillace L, Costantino C, Tomasello F, Mauro GL, Stillo M, Perrone P, Resi D, Gori D, Vitale F, Fantini MP. Attitudes toward the SARS-CoV-2 and Influenza Vaccination in the Metropolitan Cities of Bologna and Palermo, Italy. Vaccines (Basel). 2021 Oct 18;9(10):1200. doi: 10.3390/vaccines9101200.
  2. Domnich A, Grassi R, Fallani E, Spurio A, Bruzzone B, Panatto D, Marozzi B, Cambiaggi M, Vasco A, Orsi A, Icardi G. Changes in Attitudes and Beliefs Concerning Vaccination and Influenza Vaccines between the First and Second COVID-19 Pandemic Waves: A Longitudinal Study. Vaccines (Basel). 2021 Sep 13;9(9):1016. doi: 10.3390/vaccines9091016.

I suggest also the authors to reform the title of characteristics of the second row of the table 1 (short title).

Author Response

Dear authors

I’m thankful to review the paper entitled “Influenza Vaccination Uptake in the General Italian Population during the 2020-2021 Flu Season: Data from the EPICOVID 3 Online Web Based Survey” for Vaccines MDPI Journal.

My decision is Accept after minor revision.

The paper is interesting and deal with the emerging topic. The introduction is clear and well arranged. The methodology sounds good and the statistical analyses and study design are readable and informative. The discussion is good even could be improved.

 The findings of the manuscript with a large sample of participants enhance the manuscript.

We thank the reviewer for his/her suggestions that helped us to improve our manuscript.

I suggest enhancing the discussion paragraph debating the outcome of other researches detecting the vaccination coverage in Italy comparing first and second wave pandemic with follow references.

 Montalti M, Di Valerio Z, Rallo F, Squillace L, Costantino C, Tomasello F, Mauro GL, Stillo M, Perrone P, Resi D, Gori D, Vitale F, Fantini MP. Attitudes toward the SARS-CoV-2 and Influenza Vaccination in the Metropolitan Cities of Bologna and Palermo, Italy. Vaccines (Basel). 2021 Oct 18;9(10):1200. doi: 10.3390/vaccines9101200. Domnich A, Grassi R, Fallani E, Spurio A, Bruzzone B, Panatto D, Marozzi B, Cambiaggi M, Vasco A, Orsi A, Icardi G. Changes in Attitudes and Beliefs Concerning Vaccination and Influenza Vaccines between the First and Second COVID-19 Pandemic Waves: A Longitudinal Study. Vaccines (Basel). 2021 Sep 13;9(9):1016. doi: 10.3390/vaccines9091016.

We thank the reviewer for the observation and we added (discussion section) and further discussed the above mentioned references.

I suggest also the authors to reform the title of characteristics of the second row of the table 1 (short title).

We thank the reviewer for the observation and accordingly we modified “biological sex” into “sex at birth”. The short title of all Tables has been reworded.

Reviewer 2 Report

In this interesting manuscript, the authors aimed to assess and compare the influenza vaccine uptake during the 2019/2020 and 2020/2021 flu seasons in a self-selected sample of respondents to the web-based EPICOVID-19 survey and to identify the factors associated with getting an influenza vaccination.

This is an important topic as understanding factors associated with a greater willingness to get vaccinated will likely be helpful to achieve high vaccination coverage rates (VCR) in the future, when influenza activity will increase again. 

Few comments are made below:

  • the introduction is clear and well written, it highlights important points as the rise in influenza vaccination coverage in Italy; my only comment here is related to lines 69-72. Here, while introducing the increase in influenza VCR in Europe and Italy, the authors mentioned the supply shortage that may have influenced the first phase of influenza vaccination campaign in many European countries. Without further explanation, this may lead to misunderstandings, as - despite the supply shortage - influenza VCR increased in most countries during the last winter [1]. I therefore suggest mentioning the increase in VCRs in European countries, specifying that this happened despite a supply shortage that characterised the first part of their vaccination campaigns [2] and making this result even more important. 
  • In the discussion (lines 274-275) the authors state that "vaccination coverage observed in those aged ≥65 years in our survey was much higher compared to the Italian official data (79.1% vs. 65.3%, respectively) and suggests a likely selection of respondents with a higher health literacy when compared to the Italian general population". Considering that the levels of health literacy in this sample were not measured, I think the authors should better explain and discuss the reasons why they think that - among many other factors - it's the higher level of health literacy that influenced their results. 
  • In the discussion (lines 294-297) the authors mention the reasons why - in their study - the most educated were more likely to get vaccinated. I would like to ask if they considered the potential role of digital health literacy and different information seeking behaviours and how this could have influenced this specific association [3]. In particular, if the authors think that changes in digital health literacy during the last decade might have brought - in highly educated individuals - to a better knowledge about influenza vaccination and therefore to a higher uptake. 
  • Methods are appropriate and results are well described

1. Del Riccio M, Lina B, Caini S, Staadegaard L, Wiegersma S, Kynčl J, Combadière B, MacIntyre CR, Paget J. Letter to the editor: Increase of influenza vaccination coverage rates during the COVID-19 pandemic and implications for the upcoming influenza season in northern hemisphere countries and Australia. Euro Surveill. 2021 Dec;26(50). doi: 10.2807/1560-7917.ES.2021.26.50.2101143. PMID: 34915972.

2. https://www.vaccinestoday.eu/stories/flu-vaccine-shortages-time-to-prioritise/

3. Dadaczynski K, Okan O, Messer M, Leung AYM, Rosário R, Darlington E, Rathmann K. Digital Health Literacy and Web-Based Information-Seeking Behaviors of University Students in Germany During the COVID-19 Pandemic: Cross-sectional Survey Study. J Med Internet Res. 2021 Jan 15;23(1):e24097. doi: 10.2196/24097. PMID: 33395396; PMCID: PMC7813561.

Author Response

In this interesting manuscript, the authors aimed to assess and compare the influenza vaccine uptake during the 2019/2020 and 2020/2021 flu seasons in a self-selected sample of respondents to the web-based EPICOVID-19 survey and to identify the factors associated with getting an influenza vaccination.

This is an important topic as understanding factors associated with a greater willingness to get vaccinated will likely be helpful to achieve high vaccination coverage rates (VCR) in the future, when influenza activity will increase again. 

We thank the reviewer for his/her suggestions that helped us to improve our manuscript.

Few comments are made below:

  • the introduction is clear and well written, it highlights important points as the rise in influenza vaccination coverage in Italy; my only comment here is related to lines 69-72. Here, while introducing the increase in influenza VCR in Europe and Italy, the authors mentioned the supply shortage that may have influenced the first phase of influenza vaccination campaign in many European countries. Without further explanation, this may lead to misunderstandings, as - despite the supply shortage - influenza VCR increased in most countries during the last winter [1]. I therefore suggest mentioning the increase in VCRs in European countries, specifying that this happened despite a supply shortage that characterised the first part of their vaccination campaigns [2] and making this result even more important. 

We thank the reviewer for the observation and we modified the sentence in the introduction section accordingly.

  • In the discussion (lines 274-275) the authors state that "vaccination coverage observed in those aged ≥65 years in our survey was much higher compared to the Italian official data (79.1% vs. 65.3%, respectively) and suggests a likely selection of respondents with a higher health literacy when compared to the Italian general population". Considering that the levels of health literacy in this sample were not measured, I think the authors should better explain and discuss the reasons why they think that - among many other factors - it's the higher level of health literacy that influenced their results. 

We thank the reviewer for the observation. We have further discussed this topic in the discussion section. In particular, although health literacy was not measured in out questionnaire, it could be speculated that a higher health literacy could be present in our sample considering the high number of health care workers among respondents (3,523 (8.5%)). This finding highlights a possible selection bias of health care works. In fact, according to the 2019 ISTAT estimates there were 753,836 health care workers in Italy out of 59,730,000 residents accounting for 1.26% of the Italian resident population [http://dati.istat.it/Index.aspx?DataSetCode=DCIS_PERS_SANIT Accessed 4 February 2022]. In addition, it was likely that the respondents with a higher education level and a consequent possible higher health literacy were more prone to fulfil the second survey considering that high education level was overrepresented among those who allowed to participate in the second survey when compared to those who did not (27,158 (65.5%) vs 91,010 (57.8%), p<0.0001; respectively) [Adorni et al Int. J. Environ. Res. Public Health 2022, 19, 1274. https://doi.org/10.3390/ijerph19031274].

 In the discussion (lines 294-297) the authors mention the reasons why - in their study - the most educated were more likely to get vaccinated. I would like to ask if they considered the potential role of digital health literacy and different information seeking behaviours and how this could have influenced this specific association [3]. In particular, if the authors think that changes in digital health literacy during the last decade might have brought - in highly educated individuals - to a better knowledge about influenza vaccination and therefore to a higher uptake. 

We do agree with the reviewer observation and we added a sentence in the discussion section.

  • Methods are appropriate and results are well described

We thank the reviewer for his/her kind words

  1. Del Riccio M, Lina B, Caini S, Staadegaard L, Wiegersma S, Kynčl J, Combadière B, MacIntyre CR, Paget J. Letter to the editor: Increase of influenza vaccination coverage rates during the COVID-19 pandemic and implications for the upcoming influenza season in northern hemisphere countries and Australia. Euro Surveill. 2021 Dec;26(50). doi: 10.2807/1560-7917.ES.2021.26.50.2101143. PMID: 34915972.
  2. https://www.vaccinestoday.eu/stories/flu-vaccine-shortages-time-to-prioritise/
  3. Dadaczynski K, Okan O, Messer M, Leung AYM, Rosário R, Darlington E, Rathmann K. Digital Health Literacy and Web-Based Information-Seeking Behaviors of University Students in Germany During the COVID-19 Pandemic: Cross-sectional Survey Study. J Med Internet Res. 2021 Jan 15;23(1):e24097. doi: 10.2196/24097. PMID: 33395396; PMCID: PMC7813561.

Reviewer 3 Report

The authors have conducted a Survey-based approach to compare the influenza vaccine uptake in citizens age between 60-65 and >65 estimate the change and also to determine the risk factors associated with the uptake of influenza vaccination in those targeted population during 2019-2021 fall into before and during the COVID-19 pandemic. The study was designed and executed well and almost all measurable parameters were included in the questionaire. The data were analyzed appropriately and results were depicted well on the manuscript. The discussion part is well rounded however, the authors did not discuss in detail about the fear of COVID-19 in the increase of vaccine uptake during the pandemic. The authors have concluded that education and economic status are directly related to the vaccine uptake. In general similar studies are usually conducted in low- and middle-income countries than a high-income country like Italy. The findings are although not ground breaking it would definitely help the authorities to come up with better plans to promote the vaccination campaign on the target population. 

I would expect that the authors should discuss about how the fear of COVID-19 had made any changes on their decision making to get the vaccine shot. The study also do not discuss about the impact of Influenza on the mortality and morbidity in Italy. Which is critical to determine why these kind of studies are important. The authors also need to discuss the type of vaccines and subtypes of Influenza viruses circulating in Italy. Discussion of the impact of influenza on those targeted populations especially the low-educated and poor population should also be discussed. If the population is not affected severely they may not notice the disease and may not seek for treatment or vaccination.

Author Response

The authors have conducted a Survey-based approach to compare the influenza vaccine uptake in citizens age between 60-65 and >65 estimate the change and also to determine the risk factors associated with the uptake of influenza vaccination in those targeted population during 2019-2021 fall into before and during the COVID-19 pandemic. The study was designed and executed well and almost all measurable parameters were included in the questionaire. The data were analyzed appropriately and results were depicted well on the manuscript. The discussion part is well rounded however, the authors did not discuss in detail about the fear of COVID-19 in the increase of vaccine uptake during the pandemic. The authors have concluded that education and economic status are directly related to the vaccine uptake. In general similar studies are usually conducted in low- and middle-income countries than a high-income country like Italy. The findings are although not ground breaking it would definitely help the authorities to come up with better plans to promote the vaccination campaign on the target population. 

We thank the reviewer for his/her suggestions and criticism which helped us to improve our manuscript.

I would expect that the authors should discuss about how the fear of COVID-19 had made any changes on their decision making to get the vaccine shot.

We thank the reviewer for the observations. The topic was further discussed in light of our previous findings reported by our group during the first part of EPICOVID survey (first surge of the pandemic) in which we found that more fear of being infected with SARS-CoV-2 was observed in vaccinated people (both influenza and pneumococcal infection), both for themselves and for family members, confirming the possible role of fear in promoting preventive attitudes (Int. J. Environ. Res. Public Health 202118, 3248. ).

The study also do not discuss about the impact of Influenza on the mortality and morbidity in Italy. Which is critical to determine why these kind of studies are important.

We thank the reviewer for the observation. We expanded the topic in the introduction section and the following sentences were added “The impact of different flu seasons on morbidity and excess mortality is variable and, for Italy, there are still gaps in existing data. Nevertheless, there is evidence of the significant burden that influenza places each year especially on high-risk groups (Influenza Other Respir Viruses. 2022 Mar;16(2):351-365.). In addition, Italy is particularly prone to be affected by the threat of influenza considering the composition of its population with a significant proportion of people aged above 75 years that constitute a large proportion of frail people (Int J Infect Dis. 2019 Nov;88:127-134.).”

The authors also need to discuss the type of vaccines and subtypes of Influenza viruses circulating in Italy.

We thank the reviewer for the observation and we further discuss this topic and we added this sentence in the limitation section:

“The influenza virus circulation in Italy was almost suppressed during the 2020/21 flu season disregarding the concerns of a potential cocirculation of influenza and SARS-CoV-2. In particular, no influenza viruses have been detected by the national surveillance system for the first time in the last twenty-one years, with a seasonal flu epidemic which has not been established in Italy. Although seasonal influenza vaccination coverage has increased compared to previous seasons, several other factors related to COVID-19 containment such as social physical distancing, an immune memory effect and a potential viral competition could partially explain the suppression of the 2020/21 flu season (Boll Epidemiol Naz 2021; 2(2):1-6. ).”

Discussion of the impact of influenza on those targeted populations especially the low-educated and poor population should also be discussed. If the population is not affected severely they may not notice the disease and may not seek for treatment or vaccination.

We thank the reviewer for the observation. Nevertheless, we only partially agree with the speculation that more severely affected people are also those that more frequently notice the disease and seek vaccination. In particular, the influenza burden is mainly indirect and the advantage of a wide flu vaccination campaign targeting marginalized population is mainly related to the overall reduction of the burden of the diseases in the population irrespectively of education and poor economic status.

Reviewer 4 Report

Thank you for providing me with the opportunity to review this manuscript. This manuscript "Influenza Vaccination Uptake in the General Italian Popula
tion during the 2020-2021 Flu Season: Data from the EPICOVID 3
Online Web Based Survey" is a comprehensive work and presented detailed data of influenza vaccine update in the 2020-2021 Flu season data. I want to see some revisions to the manuscript prior to its acceptance.

  1. The tables and figures quality can be improved.
  2. Figures can be made clearer so that can be easily visualized. 
  3. The introduction of the manuscript can be improved by adding data that can support the title of the manuscript. 
  4. Further, I would like to compare data with those previously available data so that to draw a better picture.

Author Response

Thank you for providing me with the opportunity to review this manuscript. This manuscript "Influenza Vaccination Uptake in the General Italian Popula tion during the 2020-2021 Flu Season: Data from the EPICOVID 3 Online Web Based Survey" is a comprehensive work and presented detailed data of influenza vaccine update in the 2020-2021 Flu season data. I want to see some revisions to the manuscript prior to its acceptance.

We thank the reviewer for his/her suggestions that helped us to improve our manuscript.

  1. The tables and figures quality can be improved.

We have reworded the capitation of all the Tables. In addition, new Figures have been provided.

  1. Figures can be made clearer so that can be easily visualized. 

We have provided a new version of the Figures. The full images are attached as ZIP files in the supplementary materials.

  1. The introduction of the manuscript can be improved by adding data that can support the title of the manuscript. 

We thank the reviewer for the observation we have included a sentence with relative references regarding the increase in influenza vaccine uptake observed in Europe during the 2020/21 flu season and further discuss the effect on influenza vaccines’ stocks reduction.

  1. Further, I would like to compare data with those previously available data so that to draw a better picture.

We thank the reviewer for the observation. We have further discussed our estimates regarding influenza vaccine uptake compared to that of other Italian studies assessing previous influenza campaigns and the 2020/21 flu vaccine campaign. In addition, the problem of health literacy and the possible selection bias toward more educated respondents in the present survey have been further discussed.

Round 2

Reviewer 4 Report

Accept